# Educational innovation as a communication strategy in palliative care: A study protocol and preliminary results

**Ana Paula Salas Moreira**[1☯¤*], **Beatriz Gómez Baceiredo**[1,2☯], **Carlos Centeno**[1☯], **Carla Reigada**[1☯]

**1** ATLANTES Global Observatory of Palliative Care, Culture and Society Institute, University of Navarra, Pamplona, Spain, **2** Journalism Department, School of Communication of the University of Navarra, Pamplona, Spain

☯ These authors contributed equally to this work.
¤ Current address: Culture and Society Institute, Campus Universitario, Pamplona, Spain
* asalasmo@unav.es

**Data Availability Statement:** All relevant data are within the manuscript and its Supporting information files.

## Abstract

### Introduction

Society associates palliative care with "death" or "end of life", which cause them fear and anxiety. In Spain, the media worsens the misunderstanding by depicting a wrong picture of palliative care. Educational innovation for university students may serve as an alternative communication strategy. Care and Society is a university course designed by and for students from non-health degrees to help disseminate the palliative care message. The first year of the Teach-Inn Pal project aims to evaluate the effects of the course and to identify areas of improvement.

### Objective

To present an evaluation to determine if the course can work as a campaign to refocus the public opinion on palliative care and share the preliminary results of the pilot study.

### Methodology

A prospective Participatory Action Research study. University students enrolled in the course (n = 29) are invited to test and redesign the palliative care message. Knowledge and empathy will be measured throughout the learning process. Afterwards, qualitative, thematic, inductive analysis of the course material will be carried out. This study is registered on the ISRCTN Registry under the name "Can a university course help communicate palliative care?" (Registration number: ISRCTN10236642).

### Discussion

This study is part of a doctoral thesis. Education is used as a creative outlet, allowing rapid testing of multiple tools to create ambassadors of palliative care that may reframe the public opinion.

**Funding:** Support for this work was received from the University of Navarra. The funder provided support in the form of salaries for authors [APS, BG, CC, CR], but did not have any additional role in study design, data collection and analysis, decision to publish, or preparation of the manuscript.

**Competing interests:** The authors have declared that no competing interests exist.

## Conclusion

The understanding of students about palliative care changed, the overall impression of the experience was positive, and students were also able to explain palliative care to people with little or no experience in the topic. However, to determine if they became ambassadors the results of the mid-term assessment are required.

## 1. Introduction

Palliative care is a worldwide necessity: more than 56.8 million people need specialized care for an incurable disease each year [1]. In response to this reality, palliative care started in 1967, when Cicely Saunders founded the first hospice dedicated to providing an integral support (physical, emotional, social, and spiritual) to terminally ill patients to improve their quality of life [2, 3]. However, at present, the supply of palliative care is still not sufficient to meet the demographic demand [4, 5].

The European Association of Palliative Care (EAPC) suggested a minimum of two specialized palliative care services for every 100.000 inhabitants [6]. And yet, there are few countries that reach this threshold. From the 49 countries analyzed in the EAPC Atlas of Palliative Care in Europe, the only country that met this recommendation was Austria, with 2.2 palliative care specialized services [6]. Yet, the rest of the countries fell short on this requirement: on average, there only were 0.8 specialized services per 100,000 inhabitants [6]. As for the education, palliative care is increasingly being included in the curricula across European Medical and Nursing Schools. However, the percentage of Schools teaching palliative care per country and the way it is taught varies greatly [6]. Only 9 countries report teaching palliative care as a mandatory subject, and very few countries report offering over 20 palliative care teaching hours and mandatory clinical practice [6].

More specifically, in Spain there are 0.6 specialized services per 100,000 inhabitants [6]. And even if more than half of the Medical and Nursing Schools offer subjects related to palliative care, only 48% of the Nursing Schools and 10% of the Medical Schools include these studies as a part of their obligatory curricula [6].

Furthermore, palliative care has a communication problem, as 54 years after its creation the concept is still misunderstood. Palliative care is currently perceived as "death" or "end of life", and that misconception continues to generate feelings of bereavement, fear, and anxiety in society [7, 8]. In Spain, the media may play an important role in this misunderstanding. An analysis of 600 Spanish newspapers (2009–2014) concluded that information on palliative care does not reflect its true nature, as the narratives used to talk about it are based on ideological and moral content such as the debate of issues like euthanasia [9]. The importance of this study stems from the need to find alternative routes to reach an alienated audience. Society receives a distorted version of the palliative care message and identifying a more direct channel of communication could be a solution.

## 2. State of the art

### 2.1 Educational innovation strategy

Health-related issues are frequent in the public opinion sphere. Therefore, the possibilities offered by communication for the development of a healthier society have been studied for years [10, 11]. Due to its evolution and persuasive capacity, communication has become an essential component of health prevention and health promotion programs to inform and educate citizens [10, 12, 13]. Within the health sector, communication campaigns facilitate social

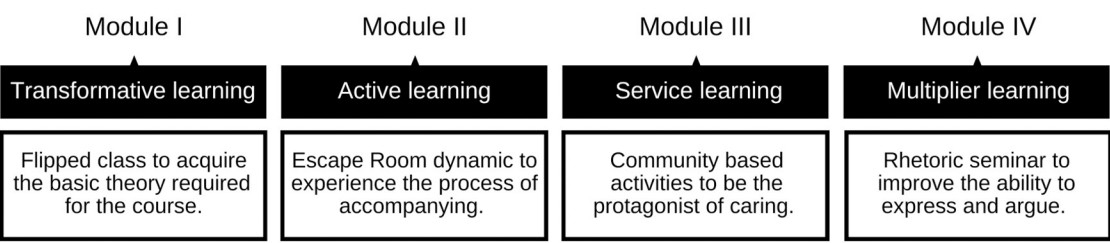

**Fig 1. Course structure diagram.** You can find further information on the Modules on section 2.3.

education because they seek to increase the knowledge to encourage new behaviors [10, 12, 14, 15]. And these campaigns have secondary goals, such as bringing attention to an issue to create an agenda [14, 16, 17].

However, the benefits also work the other way around. Educational programs such as Massive Online Open Courses (MOOC), despite not being an explicit communication campaign, have successfully started conversations around several topics and helped clarify issues [18]. MOOCS have also been widely used to disseminate research results beyond the academic sphere [19]. Specifically, the Dying2Learn MOOC demonstrated that the educational ambiance was favorable to increase participants' comfort to discuss difficult issues such as death, to foster conversations about this topic, and to explore unheard perspectives from people outside the healthcare context [20]. These results can be a foundation for the development of palliative care resources and information context [20]; which, in turn, could be the basis for formal communication campaigns.

On that line, some social studies have focused on finding new ways to overcome the confusion generated around palliative care [21, 22]. The last diagnostic study of the EnPositive-PAL project conducted with university students all over Spain highlighted that it is positive to involve students when designing a strategy to disseminate the palliative care message [23]. It used the Participatory Action Research method, Design Thinking and adopted a strategy of public engagement in Responsible Research and Innovation with the idea of creating a social intervention to promote a transformative, active, and service-learning approach [23]. The result of this process is Care and Society: a four-module course (Fig 1) aimed to create a meaningful learning experience for Spanish university students from degrees unrelated to the health sector.

## 2.2 Why university students from degrees unrelated to the health sector

In a health communication campaign, a desirable outcome is for the target audience to become an ambassador of the cause. Ambassadors, otherwise known as influencers, are people that can exert interpersonal influence or help reform the environmental condition that shapes the behaviors of the segment to be changed [24]. Creating ambassadors have two advantages. Firstly, they can customize the message to the needs and values of their peers [24]. Peer to peer communication has been proven to facilitate health messaging, because fellow community members are both trusted and relatable, and able to express information in an understandable and significant way [25]. Secondly, ambassadors are likely to respond to negative behaviors of those they are attempting to influence [24]. This type of engagement provides opportunities for giving instant feedback, having open dialogues, and answering questions immediately [26].

Moreover, as the issue of palliative care in the academic curricula of health-related degrees is already being tackled [27, 28], the aim of this course is to reach students belonging to areas of expertise outside the sector. Considering the two previous advantages, students with

different backgrounds and areas of expertise are an ideal group to create ambassadors, as the variety of backgrounds will allow the palliative care message to be disseminated beyond the field of health. The social intervention proposes to reach the students through the Core Curriculum: an initiative designed to offer students from a wide range of degrees an integral education in core areas of study [29, 30]. Aligned with that objective, the Core Curriculum course Care and Society seeks to convey to the university students the value of caring for others within society.

## 2.3 Care and Society course structure

**2.3.1 Module I: Transformative learning.**  In this module, through flipped classes, the students will receive the theory required to face the rest of the course. The students will learn to identify situations of serious illness that involve intense suffering, and to outline a set of actions that help to care for the person suffering. During this segment, the total of students will be divided in three groups to encourage participation and to adapt the class to the necessities of everyone. The groups will switch professors every two weeks, as each professor will oversee the explanation of a set of topics related to one of three categories: cancer, dementia, and neurodegenerative illnesses.

**2.3.2 Module II: Active learning.**  Through a self-learning circuit inspired on an Escape Room dynamic, students will be the protagonists in building their own knowledge. By the end of the module, the students will be able to understand what the process of assisting, caring and alleviating suffering is like. The total of students will be divided in groups of six to facilitate and encourage the interaction of students with the environment.

**2.3.3 Module III: Service learning.**  Students will participate individually in community service activities, allowing them to link what they have learned in modules I and II with the local context. By the end of the module, the student will have developed their personal and professional competencies, and will be able to evaluate, implement, and recommend good practices when caring for people.

**2.3.4 Module IV: Multiplier learning.**  All students will attend a four-hour seminar titled "An introduction to rhetoric" that will give them some resources to improve their ability to express themselves and construct valid arguments on a topic. By the end of the module, the students will be able to plan a persuasive speech according to the basics of rhetoric. The seminar will be led by a teacher specialized in communication and will be taught to all students together.

## 3. Method

This pilot study, of the three-year multicenter project Teach-Inn Pal, is part of a doctoral thesis. The project started on November 2021 and will continue until September 2024 in Spain. The pilot study took place on the second semester of the academic course 2021–2022, starting on January 2022 (Fig 2). However, enrollment to the course took place five months before, in August of 2022, when the university course registration period was open.

### 3.1 Objective

Using an interdisciplinary approach, Care and Society focuses on promoting the knowledge of palliative care and increasing the empathy of students on issues related to accompanying and caring for people suffering from a serious illness. The objective of this protocol is to present the possible evaluation of Care and Society as public health communication strategy. This proposal stems from the recognition that education is inherent to communication, and therefore

SPIRIT schedule of enrollment, interventions, and assessments

| | STUDY PERIOD | | | | | |
|---|---|---|---|---|---|---|
| | Enrollment | Allocation | Post-allocation | | | Close-out |
| Timepoint | -5 | 0 | Months 1-2 | Month 3 | Month 4 | Months 9-10 |
| **Enrollment** | | | | | | |
| Course enrollment (eligibility screen) | x | | | | | |
| Informed consents | | x | | | | |
| Allocation | | x | | | | |
| **Interventions** | | | | | | |
| Module 1: Transformative learning | | | x | | | |
| Module 2: Active learning | | | x | x | | |
| Module 3: Service learning | | | | x | x | |
| Module 4: Multiplier learning | | | | | x | |
| **Assessment** | | | | | | |
| Knowledge | | | x | x | x | |
| Empathy | | | x | x | x | |
| Communication capacity | | | | | x | |
| Knowledge retention | | | | | | x |
| Retransmission | | | | | | x |

**Fig 2. SPIRIT schedule of enrollment, interventions, and assessments.** The SPIRIT schedule is copyrighted by the SPIRIT Group and is reproduced with permission.

aims to evaluate the communicative worth of innovative education to promote a positive message of palliative care.

This research aims to address two questions:

1. Is educational innovation a suitable solution to the communication necessities of palliative care?

2. Can students help amplify the palliative care message?

## 3.2 Research design

It is a prospective study with a participatory action research framework, using multiple methods for data collection such as application of scales, satisfaction questionnaires, focus groups and direct observation. Participatory action research is composed of cyclical steps of planning, action, observation, and results, making participants evolve from the beginning [31, 32]. This allows us to receive feedback and incorporate it to improve both the course and the assessment techniques along the process before their implementation in other universities. Consequently, this is considered a pilot study, deeming the course Care and Society the prototype of the social intervention. We understand that evaluations related to the changes in attitudes and behaviors cannot be based on immediate results. Therefore, it is intended that this evaluation will be conducted over 3 years, giving us the opportunity to repeat the intervention and see its results in medium-term.

## 3.3 Participants

During the pilot, the course will be offered as an elective class to a maximum of 30 students in a Spanish university. All students who desire to participate in the course can enroll as long as

they: (1) are in their second or subsequent years of university, (2) are from the Schools of Economics and Business, Architecture, Education and Psychology, and (3) their fields are not related to health.

### 3.4 Moments of evaluation

The evaluation will assess the capacity of the course to turn students into ambassadors that can change the framework of the discussion in society. We understand as ambassadors the students who demonstrate: (1) an adequate degree of understanding of the concept of palliative care, (2) a positive attitude towards the social problem, (3) the ability to communicate the message of palliative care and (4) to retain the knowledge five months after taking the course. The overall assessment of the course will focus on four moments of evaluation (Fig 3).

**3.4.1 Moment 1: Initial assessment.** In Moment 1, one week before the beginning of the course, students will receive an email inviting them to participate in the study. The invitation will have three links: one to the initial Palliative Care Knowledge Questionnaire (PCKQ), another to a Cognitive and Affective Empathy Test (TECA), and the last to a consent form for recontact. The initial PCKQ is meant to examine participants previous knowledge on palliative care. If any student shows a formidable knowledge on the topic, they will not be excluded from the study. Their performance will simply be recorded, and any improvement will be taken into consideration.

**3.4.2 Moment 2: Learning assessment.** Following the end of Module I, students will receive links to the PCKQ and TECA of Moment 2. Furthermore, to evaluate Module II for teaching purposes, a ten-minute debriefing will be recorded at the end of the Escape Room. With their written consent, transcripts of this recording will be used to help analyze the data.

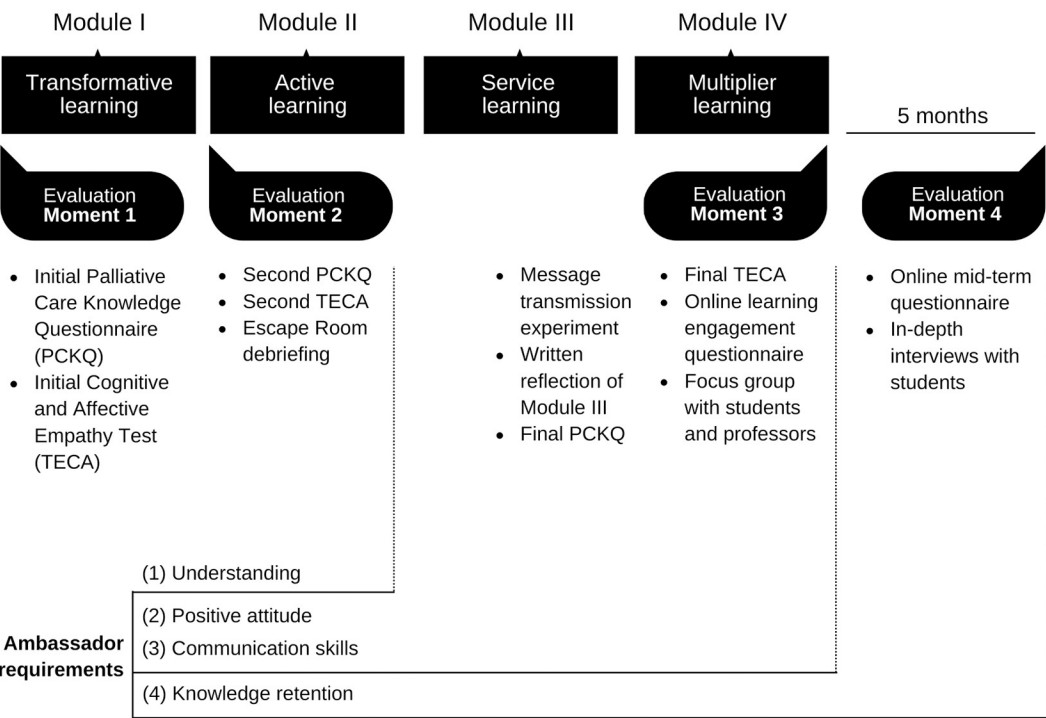

**Fig 3. Diagram of moments of evaluation.** You can find further information on the Moments of Evaluation on the following sections.

**3.4.3 Moment 3: Final assessment.** The final two classes in the course contemplate a seminar where students will learn the basics of Rhetoric. They will write a speech, applying the knowledge of palliative care acquired during the course and the resources given throughout the seminar. The speeches will be sent to and graded by the professor, who will forward them to the research team to be analyzed accordingly. The teacher will then proceed to select the five best speeches for a Message Transmission Experiment. For this experiment, five selected students will deliver their speech in front of first-year students not enrolled or related in any way to the course. Following the speeches, the first-year students will fill out a questionnaire to assess their understanding of palliative care after the experience.

Additionally, prior to the last day of class, students will turn in a written reflection regarding the Module III experience. This material will also be analyzed for the research with the prior written consent of the students. After the class is over, students will receive the link to answer the last PCKQ and TECA. Participants will also receive a link to provide feedback on the course through a satisfaction and commitment questionnaire that includes an open question and a learning commitment regarding an attitude change: "After what you have learned in this course, what do you intend to change in your attitude in the coming months to enhance the palliative care message?".

Lastly, all the students and teachers of Care and Society will be invited to participate in a focus group to evaluate the overall experience in this course. The first five students who respond to this request will be selected. The purpose of this focus group is not to evaluate the knowledge of the participants, but to discuss the strengths and weaknesses of the methodology, the content and other aspects of the course. The focus groups will later be transcribed, and thematic content analysis of the material will be carried out.

**3.4.4 Moment 4: Mid-term assessment.** Five months after the publication of course grades, students will be emailed a link to an online questionnaire to assess their mid-term knowledge and to indicate the extent to which they have implemented the learning commitment outlined in the satisfaction and commitment questionnaire. Students should respond to the question "You have set out to change your attitude to improve the palliative care message. Have you managed to accomplish this in the last few months? Please provide an example." Furthermore, they will receive an invitation to participate on in-depth interviews to talk about their experience on the months following the course and to confirm whether they have retransmitted the palliative care message at some point during that timeframe.

The 5-month timeframe was selected taking into consideration that the course grades will be made public in early June. Hence, we will be recontacting students no sooner than in November of 2022. Thus, by the time we recontact the students, some of them will be finishing their second-to-last semester in university. This is significant for two reasons. First and foremost, the recontact form they received in the Moment 1 of evaluation allowed communication via their university email. If we wait longer, some of the emails may become inoperative. Secondly, sending invitations with a semester's leeway allows us to have plenty of time to schedule interviews knowing that the students are still in the same geographic location, facilitating face-to-face interviews or at least with similar schedules to ours.

## 3.5 Data management

To protect the identity of the participants, every student will create his or her own code at the beginning of the study. The code will be the combination of a city and three numbers. It will be used in all questionnaires, so that the researcher is only be able to associate each questionnaire with a code instead of data identifying the sender.

Students will receive the first set of questionnaires in an email invitation sent through the university's in-house platform. The email addresses of the students who decide to participate will be saved by the researchers to send the next steps of the study directly to them. These emails will be stored separately and will in no way be linked to the answers of the questionnaires or to the generated codes.

The answers of the questionnaires will be analyzed after the course has been completed and the grades submitted. This guarantees that professors will not receive any information related to the study. Written consents about the contact will be destroyed once the project is finished and will not be associated in any way to the answers of the questionnaires. All information will remain confidential in accordance with the Data Protection Act.

The data from this study will be incorporated into a database or file without personal data and access will be available to the study personnel. A backup copy of the personal data file related to recruited participants will be kept on the hard drive of the principal investigator's university computer equipment and will be accessible to the principal investigator for 3 years. After those three years, all data will be deleted.

### 3.6 Ethical considerations

The study will comply at all times with the principles of the European Code of Conduct for Research Integrity. Written consents will be obtained from all the participants. The University of Navarre Research Ethics Committee has approved this study (2021.204) and will monitor that the ethical principles of research involving human subjects are adhered to.

## 4. Preliminary results

The first edition of Care and Society took place from January to May 2022. The course had 29 students enrolled. For the preliminary results, we compared the PCKQ answers of the participants throughout the course. We also assessed the results of the Message Transmission Experiment, and we obtained an initial overview of the recurring themes of the focus group.

### 4.1 Palliative Care Knowledge Questionnaire

The students were asked to complete a PCKQ in three different moments of the course: before it started (Moment 1), after the first module (Moment 2), and during the last class (Moment 3). The PCKQ included four multiple choice questions with multiselect answer options, in which students were asked to pick the options that best described palliative care, and one open question in which students were asked to define palliative care. From the 25 students that accepted to participate, 12 students responded in Moment 1, 25 responded in Moment 2 and 21 responded in Moment 3.

The analysis carried out is centered on the open question and only took into consideration respondents who answered at the 3 Moments (S1 File). Although some of the students appear to have a vague knowledge on palliative care, when asked to define palliative care in Moment 1, the answers revolved around the idea of the final moments of the patient. The definitions made use of concepts such as "terminal".

"The care and support received by terminally ill patients and their families",

P1.

"It is a treatment that is carried out in very specific and extreme circumstances to provide the patient with a state of well-being and tranquility, avoiding the pain without cure",

P2.

"Accompanying terminally ill people",

P3.

In Moment 2, we start to see a shift in perspective. While students still made use of words like "terminal" in the definitions, the answers were no longer focused on that topic. The ideas of reducing the pain and suffering, enhancing the life quality, and accompanying and caring for the patient were present in most of the answers.

"Palliative care is the support, help and assistance given to patients in advanced diseases with near-death expectation, so that they have a better quality of life and comfort, and their suffering is reduced",

P1.

"Palliative care is a treatment for chronic illnesses or those with a life expectancy of less than 6 months, which helps the patient to reduce pain and give them a better quality of life",

P2.

"Caring to reduce the suffering of a person who is terminally ill",

P3.

Finally, in the Moment 3, instead of asking for the definition one more time, the questionnaire included a variation of the same question: "What is the message you will carry with you regarding palliative care?". The answers were very diverse, but a recurrent topic was the notion of palliative care as something people should have access to: "palliative care services are indispensable", "everyone should have access to palliative care", and "we all deserve to be cared for". The PCKQ of Moment 3 also incorporated a new question: "Rate the need for training of future non-health professionals in the area of palliative care". The possible answers were a scale that went from "very necessary" to "not necessary at all". From the 21 respondents, 16 said rated palliative care knowledge as "very necessary". 4 respondents said it was "necessary". The only respondent left said it was both "necessary" and "very necessary". Thus, by the end of the course, most of the students highly regarded the training they have received, and they valued it enough to consider it necessary for other students like them.

## 4.2 Message Transmission Experiment

During the last class of the course, five students were selected to read out loud their speech in front of their classmates and two guests. The guests were two first-year students not enrolled in the course, who have previously display little or no knowledge regarding palliative care in an assignment delivered for another university course. After hearing all the speeches, they were asked to define palliative care (S2 File). One of them defined palliative care as "a medical treatment intended for patients with a terminal illness with the aim of reducing their suffering as much as possible". The other one wrote that palliative care "is a special care that focuses on people who do not have the possibility of being cured of their illness. They try to support, both focused on the disease and psychological consequences, to the patient and the family".

Two lessons can be drawn from this experience. The first one is that the students were able to learn the basics of rhetoric from the four-hour seminar. The speeches written for the

experiment were well structured and clearly expressed the concepts that the students had learnt during the course. The second one is that the speeches showed signs of being effective in conveying the concepts learnt throughout the course to the guests. The guests, who presumably knew little or nothing about palliative care prior to the experience, provided an acceptable definition of palliative care after hearing the speeches. Thus, the students of the course showed their ability to communicate and retransmit the palliative care message to their peers.

### 4.3 Focus group

Upon the course termination, we conducted a focus group with three students enrolled in the course, three of the professors of the course, and three investigators related to the field of palliative care to discuss the experience of the participants. A thematic content analysis of the focus group transcription is yet to be carried out. In the meantime, the field notes of two of the researchers in the focused group were examined to obtain an initial overview of topics discussed on the focus group (S3 File).

Among the points that students highlighted during the conversation, three stood out: (1) the number of people per group, (2) the feelings aroused by the classes, and (3) the usefulness of the material learnt. Two of the students that attended the focus group agreed that the size of the class was crucial for them to feel more comfortable speaking about the issues discussed during the course. This claim was also supported by two of the professors. Furthermore, the three students mentioned that the topic of the class and the class material generated a wide variety of feelings. There were also some comments about the importance of managing said feelings and giving the students resources to cope with those emotions as well. And, finally, all of them found the content of the class useful and applicable to their day-to-day life. An emphasis was made on the communicational aspects of the syllabus and the practical guidelines for caring for other people.

Regarding the input from the professors, the perspectives were more diverse as their tasks within the course were different. Two of them gave theory classes in the Module 2, where the groups where smaller; and the other one gave the Module 4 class with the full class. Yet, there were some areas of common ground. The three professors found this course to be challenging, either because of the difficulties in communicating to an audience unrelated to the health sector or because of the complexity of the content itself. Additionally, a recurrent topic during the conversation was the diversity of profiles among the students, as they all came from different degrees. In relation to this diversity, two of the professors mentioned that one of the things they enjoyed most of the experiences was reading the students insights, thoughts, and reflections. Lastly, two of the professors also pointed out that the course evoked a feeling of a shared experience with the students.

## 5. Discussion

In the first year of the Teach-Inn Pal project, the protocol proposed an assessment method to evaluate the effectiveness of the course in transmitting the palliative care message to people beyond the health sector. This protocol also assessed the course capacity to create ambassadors of palliative care who can generate new frameworks for dialogue within society. To respond to these propositions, we identified two research questions that will be discussed below.

### 5.1 Research question 1: Is educational innovation a suitable solution to the communication necessities of palliative care?

We believe that this approach is innovative as it averts the usual health promotion programs, in which the information moves on a one-way flow from the sender to the recipients and is

heavily dependent on the media as a channel [9, 24]. This social intervention shifts the trajectory in the dissemination of palliative care knowledge to society. Rather than transferring knowledge on a hierarchical basis, it is a designed to be an extended experience that allowed students to witness what care looks like and have dialogues with professionals dedicated to caring.

By doing so, not only do we involve the students in their own learning, but we also provide them with the opportunity to redirect and adapt the syllabus to fit their needs. We thus ensure that students value their learning, which may result in said learning becoming perdurable and worthy of being retransmitted. Educational innovation seems to be a tailored solution to the communication necessities of palliative care, as it strives to leave a long-lasting impression on students, ensuring that the palliative care message is well embedded on the recipients.

### 5.2 Research question 2: Can students help amplify the palliative care message?

This project perceives students of the Core Curriculum who are not in the health sector as potential re-transmitters of the palliative care message to society. If the information learnt through the course resonates with the students, they can potentially serve as a conduit to amplify the palliative care message as a result of the students' high connectivity [24, 33]. High connectivity, or the ability to bridge multiple groups because of the formation of weak-tie relationships, allow the students to take what they have learnt in the course and share it with a wide variety of different audiences [33, 34]. Not only they may share it with other students in their degrees who didn't enroll in this optative, but also with their family, friends and acquittances outside of the university environment.

Thus, we believe Care and Society used education as creative outlet that utilized rapid testing of multiple tools to create palliative care ambassadors in society. And we are also confident that these ambassadors are potential seeds of change, as they will be able both to send the message to multiple recipients and to react directly and immediately if they witness an incorrect portrayal of palliative care in their day-to-day life [25, 35]. This might potentially change the public opinion on palliative care in the long run.

## 6. Limitations

When launching the pilot, we decided not to have a control group. The two main reasons were: we did not consider the experience of a student taking Care and Society to be comparable to that of a student taking any other subject of the Core Curriculum offerings. And, secondly, the questionnaires themselves could lead to bias. By taking the same questionnaire repeatedly, it could generate curiosity in the control group participants, who might be able to acquire palliative care knowledge on their own and improve their performance in the following questionnaires. Nonetheless, for future attempts, we believe that the existence of a control group is advisable to rule out external change factors, such as exposure to news about the topic or to other media outlets.

The code system is another variable to take into consideration. As there is a time frame superior to a month in between moments of evaluation, some of the participants had a hard time remembering the city and the numbers they chose at the beginning of the pilot study. Furthermore, some of the participants chose the same city and the same combination of numbers. As a result, some of the answers had to be excluded from the preliminary results. In future attempts, a new system of codes might be helpful to improve the data recollection.

## 7. Conclusions

The preliminary results of the pilot, delivered to 29 students from degrees not related to health sciences, suggest that the understanding of the students about palliative care changed and some of the common misconceptions were clarified. Additionally, the overall impressions of the experience were positive. Both students and professors enjoyed the approach of the course and found its content useful and valuable for their day-to-day life. After completing the course, some of the students appeared to be able to explain what palliative care is to guest students with little or no experience in the topic. Nonetheless, in future evaluations more work needs to be done in assessing the efficiency of the message transmission. A more accurate assessment of the recipients' previous knowledge on the topic needs to be carried out.

Furthermore, to evaluate if these students became ambassadors of palliative care an in-depth analysis of the course material and the results of the mid-term assessment are required. Up to now, the results hint that: (1) the students have improved their knowledge on palliative care, (2) they share a positive attitude regarding palliative care, and (3) they have started to communicate palliative care to their peers. But the mid-term assessment will show more definite results in the previous three aspects. In addition, it will provide us with insights regarding the permanence of the learning and the initiative of the students to advocate for palliative care if faced with a situation in which palliative care is being wrongly depicted. Further analysis is also needed to evaluate if the course generates a change in the levels of empathy. Moreover, the benefits of the different education strategies used in this course need to be studied in more detail.

## Supporting information

**S1 Checklist. SPIRIT checklist.** The SPIRIT checklist is copyrighted by the SPIRIT Group and is reproduced with permission.
(PDF)

**S1 Protocol. Protocol approved by the ethics committee.** Approved on February 17, 2022.
(PDF)

**S2 Protocol. Translation of the protocol approved by the ethics committee.** Approved on February 17, 2022.
(PDF)

**S1 File. Palliative Care Knowledge Questionnaire answers.** Answers with codes that appear in at least two moments of evaluation were included. Answers with codes that appear only in one moment of evaluation or answers with repeated codes in the same moment of evaluation were excluded.
(PDF)

**S2 File. Message Transmission Experiment answer sheets.**
(PDF)

**S3 File. Script and field notes of the focus group.**
(PDF)

## Author Contributions

**Conceptualization:** Ana Paula Salas Moreira, Beatriz Gómez Baceiredo, Carlos Centeno, Carla Reigada.

**Methodology:** Ana Paula Salas Moreira, Carla Reigada.

**Supervision:** Beatriz Gómez Baceiredo, Carlos Centeno, Carla Reigada.

**Visualization:** Ana Paula Salas Moreira.

**Writing – original draft:** Ana Paula Salas Moreira.

**Writing – review & editing:** Ana Paula Salas Moreira, Beatriz Gómez Baceiredo, Carla Reigada.

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
