## [Decision Letter · Decision Letter 0]

22 Feb 2023

PONE-D-22-33602Educational innovation as a communication strategy in Palliative CarePLOS ONE

Dear Dr. Salas Moreira,

Thank you for submitting your manuscript to PLOS ONE. After careful consideration, we feel that it has merit but does not fully meet PLOS ONE’s publication criteria as it currently stands. Therefore, we invite you to submit a revised version of the manuscript that addresses the points raised during the review process.

We look forward to receiving your revised manuscript.

Kind regards,

Samane Shirahmadi, PhD

Academic Editor

PLOS ONE

Journal Requirements:

2. During your revisions, please note that a simple title correction is required: Study protocol requires the word ""protocol' to be included in the title. Please ensure this is updated in the manuscript file and the online submission information.

a) If there are ethical or legal restrictions on sharing a de-identified data set, please explain them in detail (e.g., data contain potentially sensitive information, data are owned by a third-party organization, etc.) and who has imposed them (e.g., an ethics committee). Please also provide contact information for a data access committee, ethics committee, or other institutional body to which data requests may be sent. Please note that authors, including Corresponding Authors, are not permitted to be the sole point of contact for data requests.

b) If there are no restrictions, please provide the minimal anonymized data set necessary to replicate your study findings as either Supporting Information files or to a stable, public repository and provide us with the relevant URLs, DOIs, or accession numbers. For a list of acceptable repositories, please see http://journals.plos.org/plosone/s/data-availability#loc-recommended-repositories.

5. Please ensure that you refer to Figures 1 and 2 in your text as, if accepted, production will need this reference to link the reader to the figure.

Additional Editor Comments:

Our Initial Quality Check of your submission has now taken place. As a result, we need you to address the following points before your manuscript can progress any further:

1. The trial must be registered prior to submission of your protocol in one of the publicly accessible registries approved by the WHO or ICMJE (International Committee of Medical Journal Editors).

2. The name of the registry and the trial or study registration number must be included in the Abstract.

3. A copy of the protocol that was approved by the ethics committee must be submitted as a supplementary information file. Please provide an additional English translation if the original document is not in English. Please note that the protocol will be published with the manuscript if accepted.

4. A SPIRIT schedule of enrollment, interventions, and assessments must be included as the manuscript’s Figure 1, and a completed SPIRIT checklist must be uploaded as Supporting Information file S1.

Reviewers' comments:

Reviewer's Responses to Questions

**Comments to the Author**

1. Does the manuscript provide a valid rationale for the proposed study, with clearly identified and justified research questions?

Reviewer #1: Yes

Reviewer #2: Yes

2. Is the protocol technically sound and planned in a manner that will lead to a meaningful outcome and allow testing the stated hypotheses?

Reviewer #1: Yes

Reviewer #2: Yes

3. Is the methodology feasible and described in sufficient detail to allow the work to be replicable?

Reviewer #1: Yes

Reviewer #2: Yes

4. Have the authors described where all data underlying the findings will be made available when the study is complete?

Reviewer #1: Yes

Reviewer #2: Yes

5. Is the manuscript presented in an intelligible fashion and written in standard English?

Reviewer #1: Yes

Reviewer #2: Yes

6. Review Comments to the Author

You may also provide optional suggestions and comments to authors that they might find helpful in planning their study.

Reviewer #1: Dear researcher,

Your effort and motivation in doing this research is commendable. Please pay attention to the following points and correct them.

1- In part 3.3, why did you consider the evaluation time 5 months after the first stage? Is there a specific reference or reason?

2- In the first stage of the evaluation, what ways did you use to reduce the error in selecting the participants? For example, did you examine participants' previous knowledge or exposure to palliative care? How? Did you exclude such people from the research? If so, please mention it.

3- In part 4.2, or the focus group, did you assess the previous knowledge of 2 guests about palliative care? Because at the end of the article, you have evaluated their learning about palliative care and you have stated one of the most important results of your research on the basis that: "Our participants' knowledge about palliative care has increased and their skills in transferring This knowledge to others (guests), has been acceptable.

5- In section 4.3, although in the abstract of the article, you stated the method of data analysis of focus groups, it would be better if you mentioned this first in this part of the article.

With respect

Reviewer #2: Abstract

- Methods: determine of date and place of study.

-Findings are ambiguous and not explained.

Introduction

- In the introduction, the importance of the study and the existing gap should be explained

The discussion needs more analysis and interpretation

7. PLOS authors have the option to publish the peer review history of their article (what does this mean?). If published, this will include your full peer review and any attached files.

Reviewer #1: No

Reviewer #2: **Yes: **Farshid Shamsaei; Professor of nursing

---

## [Author Response · Author response to Decision Letter 0]

16 Mar 2023

Dear Reviewers and Editors of PLOS ONE: 

Allow me to start by extending my sincerest gratitude to the editors who have invested time and effort in finding reviewers for this paper, as well as to the reviewers who have agreed to take upon such request. It carries a lot of weight for us to know that the research we do ignites interest beyond the communication field. Thus, it is with great pleasure that we resubmit our article for further consideration. 

I would also like to thank all of you beforehand for all the feedback and suggestions. You have raised important questions and I will proceed to enumerate the answers in the following letter. 

Reviewer #1: 

Dear researcher,

Your effort and motivation in doing this research is commendable. Please pay attention to the following points and correct them.

1- In part 3.3, why did you consider the evaluation time 5 months after the first stage? Is there a specific reference or reason?

We have supplemented the section 3.4.4 with explanations as to why a 5-month time frame was selected (page 17, paragraph 2).

>>>The 5-month timeframe was selected taking into consideration that the course grades will be made public in early June. Hence, we will be recontacting students no sooner than in November of 2022. Thus, by the time we recontact the students, some of them will be finishing their second-to-last semester in university. This is significant for two reasons. First and foremost, the recontact form they received in the Moment 1 of evaluation allowed communication via their university email. If we wait longer, some of the emails may become inoperative. Secondly, sending invitations with a semester's leeway allows us to have plenty of time to schedule interviews knowing that the students are still in the same geographic location, facilitating face-to-face interviews or at least with similar schedules to ours.

2- In the first stage of the evaluation, what ways did you use to reduce the error in selecting the participants? For example, did you examine participants' previous knowledge or exposure to palliative care? How? Did you exclude such people from the research? If so, please mention it.

We have redrafted and added further information on the section 3.4.1 (page 14, paragraph 2) to establish a clearer focus.

>>>The initial PCKQ is meant to examine participants previous knowledge on palliative care. If any student shows a formidable knowledge on the topic, they will not be excluded from the study. Their performance will simply be recorded, and any improvement will be taken into consideration. 

3- In part 4.2, or the focus group, did you assess the previous knowledge of 2 guests about palliative care? Because at the end of the article, you have evaluated their learning about palliative care and you have stated one of the most important results of your research on the basis that: "Our participants' knowledge about palliative care has increased and their skills in transferring This knowledge to others (guests), has been acceptable.

Regarding the section 4.2 on the Message Transmission Experiment, we have elaborated on the assessment of the previous knowledge (page 23, paragraph 2) and we also acknowledge in section 7 that a better assessment of previous knowledge of recipients needs to be done in future interventions (page 30, paragraph 2).

>>>1) During the last class of the course, five students were selected to read out loud their speech in front of their classmates and two guests. The guests were two first-year students not enrolled in the course, who have previously display little or no knowledge regarding palliative care in an assignment delivered for another university course. After hearing all the speeches, they were asked to define palliative care.

>>>2) After completing the course, some of the students appeared to be able to explain what palliative care is to guest students with little or no experience in the topic. Nonetheless, in future evaluations more work needs to be done in assessing the efficiency of the message transmission. A more accurate assessment of the recipients’ previous knowledge on the topic needs to be carried out.

Regarding the section 4.3 on the Focus Group, we have not done any previous knowledge assessment as our sense is that this activity did not aim to evaluate knowledge of any sort. We have added a clarification of our scope and give further information in section 3.4.3 (page 13, paragraph 2).

>>>The purpose of this focus group is not to evaluate the knowledge of the participants, but to discuss the strengths and weaknesses of the methodology, the content and other aspects of the course.

5- In section 4.3, although in the abstract of the article, you stated the method of data analysis of focus groups, it would be better if you mentioned this first in this part of the article.

We agree with your assessment, and we have reflected this comment by adding the information in section 3.4.3 (page 16, paragraph 3).

>>>The focus groups will later be transcribed, and thematic content analysis of the material will be carried out.

We have also incorporated your comments regarding the results by redrafting parts of the section 4.1 Palliative Care Knowledge Questionnaire (page 20) and section 7 Conclusions (page 30). 

Reviewer #2: 

-Methods: determine of date and place of study.

We agree with you and have incorporated this suggestion in section 3 (page 11, paragraph 2).

>>>The project started on November 2021 and will continue until September 2024 in Spain. The pilot study took place on the second semester of the academic course 2021-2022, starting on January 2022 (Fig 2). However, enrollment to the course took place five months before, in August of 2022, when the university course registration period was open. 

-Findings are ambiguous and not explained.

We have elaborated on various points throughout section 4. Preliminary results (page 20). We hope these revisions provide more clarity. 

Introduction

- In the introduction, the importance of the study and the existing gap should be explained

We have revised the text (page 5, paragraph 2) to reflect the existing gap and the importance of the study. 

>>>Furthermore, palliative care has a communication problem, as 54 years after its creation the concept is still misunderstood. Palliative care is currently perceived as "death" or "end of life”, and that misconception continues to generate feelings of bereavement, fear, and anxiety in society [7,8]. In Spain, the media may play an important role in this misunderstanding. An analysis of 600 Spanish newspapers (2009-2014) concluded that information on palliative care does not reflect its true nature, as the narratives used to talk about it are based on ideological and moral content such as the debate of issues like euthanasia [9]. The importance of this study stems from the need to find alternative routes to reach an alienated audience. Society receives a distorted version of the palliative care message and identifying a more direct channel of communication could be a solution.

-The discussion needs more analysis and interpretation

We agree and we have elaborated on several points throughout section 5. Discussion (page 21). 

-The written format of the article is not suitable in different sections. For example, the introduction is long and the discussion is short

We have now both reduced the length of the Introduction and elaborate on the Results and Discussion. We think these changes now better the balance of the different sections. We hope that you agree.

We would like to thank you one more time for allowing us to enhance our manuscript with your comments and queries. We have diligently incorporated your feedback and hope that these revisions persuade you to accept our submission.

Sincerely,

Ana Paula Salas

---

## [Editor Report · Decision Letter 1]

27 Mar 2023

PONE-D-22-33602R1Educational innovation as a communication strategy in Palliative Care: a study protocol and preliminary resultsPLOS ONE

Dear Dr.  Salas,

Thank you for submitting your manuscript to PLOS ONE. After careful consideration, we feel that it has merit but does not fully meet PLOS ONE’s publication criteria as it currently stands. Therefore, we invite you to submit a revised version of the manuscript that addresses the points raised during the review process.

ACADEMIC EDITOR: I have returned the manuscript to you to upload the correct file.

We look forward to receiving your revised manuscript.

Kind regards,

Samane Shirahmadi, PhD

Academic Editor

PLOS ONE

---

## [Author Response · Author response to Decision Letter 1]

28 Mar 2023

ACADEMIC EDITOR: I have returned the manuscript to you to upload the correct file.

We have corrected the file.

---

## [Decision Letter · Decision Letter 2]

15 May 2023

Educational innovation as a communication strategy in Palliative Care: a study protocol and preliminary results

PONE-D-22-33602R2

Dear Dr. Ana Paula Salas Moreira,

We’re pleased to inform you that your manuscript has been judged scientifically suitable for publication and will be formally accepted for publication once it meets all outstanding technical requirements.

Kind regards,

Samane Shirahmadi, PhD

Academic Editor

PLOS ONE

Reviewers' comments:

Reviewer's Responses to Questions

**Comments to the Author**

1. Does the manuscript provide a valid rationale for the proposed study, with clearly identified and justified research questions?

Reviewer #1: Yes

Reviewer #2: Yes

2. Is the protocol technically sound and planned in a manner that will lead to a meaningful outcome and allow testing the stated hypotheses?

Reviewer #1: Yes

Reviewer #2: Yes

3. Is the methodology feasible and described in sufficient detail to allow the work to be replicable?

Reviewer #1: Yes

Reviewer #2: Yes

4. Have the authors described where all data underlying the findings will be made available when the study is complete?

Reviewer #1: Yes

Reviewer #2: Yes

5. Is the manuscript presented in an intelligible fashion and written in standard English?

Reviewer #1: Yes

Reviewer #2: Yes

6. Review Comments to the Author

You may also provide optional suggestions and comments to authors that they might find helpful in planning their study.

Reviewer #1: It has made all the necessary corrections.

Reviewer #2: The article entitle "Educational innovation as a communication strategy in Palliative Care: a study protocol and preliminary results" has been revised and changes based on comments are accepted.

7. PLOS authors have the option to publish the peer review history of their article (what does this mean?). If published, this will include your full peer review and any attached files.

Reviewer #1: No

Reviewer #2: No

---

## [Editor Report · Acceptance letter]

2 Jun 2023

PONE-D-22-33602R2 

Educational innovation as a communication strategy in Palliative Care: a study protocol and preliminary results 

Dear Dr. Salas Moreira:

I'm pleased to inform you that your manuscript has been deemed suitable for publication in PLOS ONE. Congratulations! Your manuscript is now with our production department. 

Kind regards, 

on behalf of

Dr. Samane Shirahmadi 

Academic Editor

PLOS ONE